# Analyzing protein dynamics from fluorescence intensity traces using unsupervised deep learning network

Jinghe Yuan [1✉], Rong Zhao[2], Jiachao Xu[1], Ming Cheng[1], Zidi Qin[3], Xiaolong Kou[1] & Xiaohong Fang[1,3✉]

We propose an unsupervised deep learning network to analyze the dynamics of membrane proteins from the fluorescence intensity traces. This system was trained in an unsupervised manner with the raw experimental time traces and synthesized ones, so neither predefined state number nor pre-labelling were required. With the bidirectional Long Short-Term Memory (biLSTM) networks as the hidden layers, both the past and future context can be used fully to improve the prediction results and can even extract information from the noise distribution. The method was validated with the synthetic dataset and the experimental dataset of monomeric fluorophore Cy5, and then applied to extract the membrane protein interaction dynamics from experimental data successfully.

[1] Key Laboratory of Molecular Nanostructure and Nanotechnology, CAS Research/Education Center for Excellence in Molecular Sciences, Institute of Chemistry, Chinese Academy of Sciences, 100190 Beijing, China. [2] Division of Chemical Metrology and Analytical Science, National Institute of Metrology, 100029 Beijing, China. [3] University of Chinese Academy of Sciences, 100049 Beijing, China. ✉email: jhyuan@iccas.ac.cn; xfang@iccas.ac.cn

Many membrane proteins form multimeric complexes before they carry out diverse essential cellular functions[1]. Investigating their organization and dynamics is very essential for understanding these functions[2]. Photobleaching event counting, based on the irreversible and stochastic loss of fluorescence, is a powerful tool for the stoichiometry analysis of protein complexes composed of many subunits[2]. By measuring the fluorophore photobleaching process, the information on the subunit number in a complex and the photobleaching step location can be obtained[3]. However, photobleaching event counting is not able to extract the dynamic information of membrane protein interactions, such as durations of protein association, and transition rates of proteins between different aggregation states, etc. Yet, these dynamic properties are unarguably of most interest, as they underlie protein function within cells[4]. By analyzing diffusion states, these dynamics can be extracted from the diffusion tracks, and diffusion state analysis is becoming a particularly interesting tool[5–8]. Actually, the fluorescence intensity traces can be used not only to the photobleaching event counting, but also to the protein dynamic extracting by monitoring the fluorescence intensity changes (step-jump). The main difference of dynamic finding from photobleaching event counting is that the dynamic finding is to track the processes of molecule association and dissociation, which is presented as the molecule diffusion state changes or fluorescence intensity jumps.

No matter for photobleaching event counting or for dynamic finding, extracting the step-like information embedded in heavy noise from fluorescence intensity traces is still a challenging task[9,10]. When the noise amplitude is larger than the step-drop, it is very difficult to discern between step and noise. Moreover, even with higher signal-noise-ratios, filtering out the fluorophore blinking driven by core fluorophore instabilities from the true steps is also a big obstacle.

For photobleaching event counting, many methods have been developed to find the drop events from the fluorescence intensity traces. The simplest way may be to use filters such as Chung and Kennedy's nonlinear filter[11], or Haar wavelet-based filter[12,13] to reduce noise and detect the drops by observing. To improve the convenience and accuracy, more automatic approaches, including data treated with wavelet transformation[14,15], multiscale products analysis[16], running $t$-test algorithm[17], step-fitting based on $\chi^2$ distribution[18], hidden Markov models (HMM)[19,20] and other algorithms have been developed to analyze single-molecule data. Among these methods, HMM have been widely used because the discrete transitions between states can be considered as memoryless (Markovian) processes and the time context is taken into account[20]. However, all these approaches above need initial selecting parameters, which can be very subjective even unreasonable. Furthermore, in HMM methods, the memoryless hypothesis may be not practical. Even with these mathematical aids, for complexes with >5 subunits, the distributions of bleaching steps for $n$ and $n+1$ subunits look similar, and detection of discrete steps becomes more difficult[21,22].

Deep-neural networks (DNNs) have achieved excellent success on various kinds of difficult learning tasks. We also developed a combining method of using convolutional and long short-term memory (LSTM) deep-learning neural network (CLDNN) for photobleaching event counting[23]. The convolutional layers take charge of feature extraction of step-drop (photobleaching) events and LSTM recurrent layers distinguish between photobleaching and photoblinking. This CLDNN does not require user-specified parameters and gives very high accuracy. However, CLDNN still belongs to supervised-learning framework, which needs large labeled training sets of input-target pairs to train this network. In practice, it is often difficult

even impossible to label the experimental data without subjectivity. In fact, extracting step-jump events (such as photobleaching event counting or dynamic finding) can be regarded as mapping of sequences to sequences, and LSTM alone is qualified to the two tasks of extracting jump information and distinguishing photobleaching from photoblinking. After getting rid of the convolutional layers, the sate paths (hence the drop location and the transition rates between different states) can be extracted as an extra benefit.

In this work, instead of the diffusion states analysis, with the fluorescence intensity traces, we propose an unsupervised-learning framework of discriminator-generator network (DGN) to analyze the aggregation states and association dynamics of membrane proteins[24]. This framework can be used not only for photobleaching event counting, but also for the dynamic finding. This proposed DGN network consists of two bidirectional Long Short-Term Memory (biLSTM) networks[25], in which, a biLSTM as discriminator is used to map the input time trace of fluorophore intensity to a hidden state vector of a fixed dimensionality, and then another biLSTM as generator to recover the input time trace from this hidden state vector. The discriminator and generator of the proposed framework are jointly trained to maximize the conditional probability of predicting itself with the input time trace, and thus find the hidden state (fluorophore count sequence) behind the time trace.

This framework was trained in an unsupervised manner based only on raw time traces and without the need for any supervisor annotation. Unsupervised learning has two obvious benefits, one is that it avoids the prejudice of selecting training data, and another is that we need not to label the traces for training the nets, which is often difficult or even impossible.

LSTM does not suffer on very long-time trace[26], so can disregard the memoryless hypothesis in hidden Markovian model (HMM). And biLSTM can access both past and future context to improve the prediction results[25,27].

Our framework refers to three types of unsupervised-learning framework, auto-encoding variational Bayes (VAEs)[28], Generative Adversarial Nets (GAN)[29], and the translation frameworks[30,31]. VAEs method needs to build the model density, it will not work in the case where we are dealing with distributions supported by low-dimensional manifolds[32]. It has two main differences from GAN as well. The first one is that the framework of GAN consists of a generative model and a discriminative model sequentially, which core goal is to train the generative model to generate some samples based on the probability distribution, however, our framework has an inverse structure and our main goal is to optimize the discriminator to find the hidden states. The second difference is that we use recurrent neural network (RNN) model LSTM to map the time sequences but GAN used multilayer perceptron in both the generative model and discriminative model. The main difference from translation frameworks is that we used sequential hidden states instead of the summary hidden states. To the best of our knowledge, this is the first time such a system has been demonstrated and we believe our work will provide a new paradigm for dynamic analysis of membrane proteins.

This proposed framework comes with many advantages relative to previous modeling frameworks. The first one is that no Markov hypothesis needed, which is essential for the HMM methods[33–35]. The second one is that the unsupervised training results with neither pre-knowledge (such as step number) nor pre-labeling of experimental data are required[23]. Remarkably, the third one is that the embedded LSTM units can not only reduce the interferences from all types of noise, but also even extract step information from the noise distribution, which is impossible for human identifier.

## Results

**Network architecture**. We assume the fluorophore number as the hidden state, the changing process of emitting fluorophores as the hidden process, $h$, which is not necessary Markovian process. As in Fig. 1, the generative process is the process to generate the fluorescence intensity trace $x$ under the hidden process $h$. Now our most concerned task is to extract the hidden process, $h$ from the measured fluorescence intensity trace $x$, named discriminative process as in Fig. 1. Statistically, in the whole framework combining the discriminative process and generative process, it means to maximize the probability

$$P(x) = \prod_{t=1}^{T} P(x_t | x_{t-1}, \cdots, x_1), \tag{1}$$

where $t$ is the time notation and $T$ is the sequence length.

This can be done with our proposed framework of RNN discriminator-generator. We used a biLSTM layer and a softmax layer for the discriminator and a biLSTM layer and a linear layer for the generator.

As the discriminator reads each element of the input sequence $x$ sequentially, the hidden state (output of the discriminator) changes according to equation (2).

$$h_t = f(h_{t-1}, x_t). \tag{2}$$

After reading the whole input sequence, we get the hidden state sequence of the same length.

The generator of the proposed model is trained to generate the output sequence $\{\widehat{x}_1, \widehat{x}_2, \cdots, \widehat{x}_t, \cdots, \widehat{x}_T\}$ to approximate the input sequence $\{x_1, x_2, \cdots x_t, \cdots x_T\}$ by predicting the element $x_t$ with the given hidden sequence $\{h_1, h_2, \cdots, h_t\}$ and previously predicted sequence $\{\widehat{x}_1, \widehat{x}_2, \cdots, \widehat{x}_{t-1}\}$. So $\widehat{x}_t$ is also conditioned on $h_t$ and $\widehat{x}_{t-1}$, Hence, the output of the generator at moment $t$ is,

$$\widehat{x}_t = g(h_t, \widehat{x}_{t-1}). \tag{3}$$

The discriminator and generator of the proposed framework are jointly trained in an unsupervised manner to minimize quadratic loss function

$$L = \frac{1}{2T} \sum_{t=1}^{T} (\widehat{x}_t - x_t)^2. \tag{4}$$

Once the training is completed, the discriminator can be independently used to estimate the hidden sequence (fluorophore number sequence), meanwhile the generator to generate time sequence under given the hidden sequence.

**Training the network**. Just as training GANs, training DGNs is delicate and unstable as well. Inspired by the DE-GAN[36], we extracted the mean fluorescence intensity of single-fluorescent molecule from the experimental datasets with gaussian mixture model method[37] (Supplementary Methods and Supplementary Fig. 1), and then synthesized datasets with this mean fluorescence intensity and taking into consideration of the three origins of noise, namely, the Poisson distributed shot noise, Gaussian distributed random noise, and fluorescence blinking (Fig. 2 and Supplementary Methods). To describe the noise level, as in our previous report[23], we adopted the average of the adjusted signal-to-noise ratio (aSNR)[10] defined as

$$a\text{SNR} = \frac{2(\mu_{i+1} - \mu_i)}{\sigma_i + \sigma_{i+1}}, \; i = 1, 2, \cdots N - 1, \tag{5}$$

where $\mu_i$ and $\sigma_i$ are the mean and standard deviation (SD) of the $i$th fluorescence state (a fluorescence state is the plateau of data points between two steps) and $N$ is the fluorescence state number of the fluorescence time trace.

To facilitate the convergence of DGN, we enhanced the loss function by adding a hidden-space loss function (cross-entropy loss function) for the synthesized datasets

$$L_h = \frac{1}{T} \sum_{t=1}^{T} y_t \ln(h_t), \tag{6}$$

where $h_t$ is the output of the discriminator and $y_t$ is the ground truth of the synthetic data, and amended the backpropagation error of the discriminator as

$$\delta_d = (1 - \alpha)\delta_g + \alpha\delta_h, \tag{7}$$

where $\delta_g$ is the backpropagation error of experimental datasets propagating from the generator to the output layer of discriminator, and $\delta_h$ is the backpropagation error of synthesized datasets generated at the output layer of discriminator. $\alpha$ is a weighted factor

$$\alpha = 0.5e^{-\frac{2k}{K}}, \tag{8}$$

where $k$ is current training loop and $K$ is the total training loops.

To train DGN by this improved strategy, the networks is forced to carry information of the synthesized datasets at the beginning, and more information of the experimental datasets is used gradually but guaranteeing the convergence in the direction trained with the synthesized datasets.

For both photobleaching event counting and dynamic finding with the fluorescence intensity traces, we built two datasets consist of a synthesized subset and an experimental subset used for training, validating and testing the proposed framework, respectively[38].

Datasets with different signal-to-noise values were synthesized (Supplementary Fig. 2). Then this synthesized dataset was randomly divided into three subsets again, namely, the training subset, validating subset and testing subset. The experimental subsets were collected from the experimental data of actual single-molecule movies. These experimental subsets were also divided into three subsets, namely, the training subset, validating subset and testing subset. Both the synthesized and experimental training subsets together is input to the nets for training and both the synthesized and experimental validating subsets together for validation. The hypothetical maximum state number is set as 10 for photobleaching counting and 5 for dynamic finding, respectively (Fig. 2(a)).

Both direction LSTM of the discriminator include 32 LSTM units. Its parameters were optimized by minimizing cross-entropy loss function using mini-batch gradient descent with batch size 8 to maintain a balance between the robustness of stochastic gradient descent and the efficiency of batch gradient descent. Weights are randomly initialized. To reduce the overfitting and guarantee convergence, the techniques known as L2 regularization[39] and weights thresholding[40] were used.

There are 16 LSTM units in both direction LSTM of the generator. Its parameters were optimized by minimizing mean square error loss function using mini-batch gradient descent with batch size 8. Weights are randomly initialized as well. The techniques of L2 regularization and weights thresholding were used too. The parameters presented in the network are detailed in Supplementary Table.1.

**Model evaluation**. We trained and validated this DGN networks as in Fig. 2(a). Supplementary Fig. 3 shows the loss curves for training and validation, respectively. After 60 epochs, the training and validation loss is 0.033 and 0.058, respectively, for photobleaching event counting, and after 34 epochs, the training and validation loss is 0.026 and 0.15, respectively, for dynamic finding. The validation loss is larger than the training loss and going down uniformly, indicating that the overfitting had not appeared.

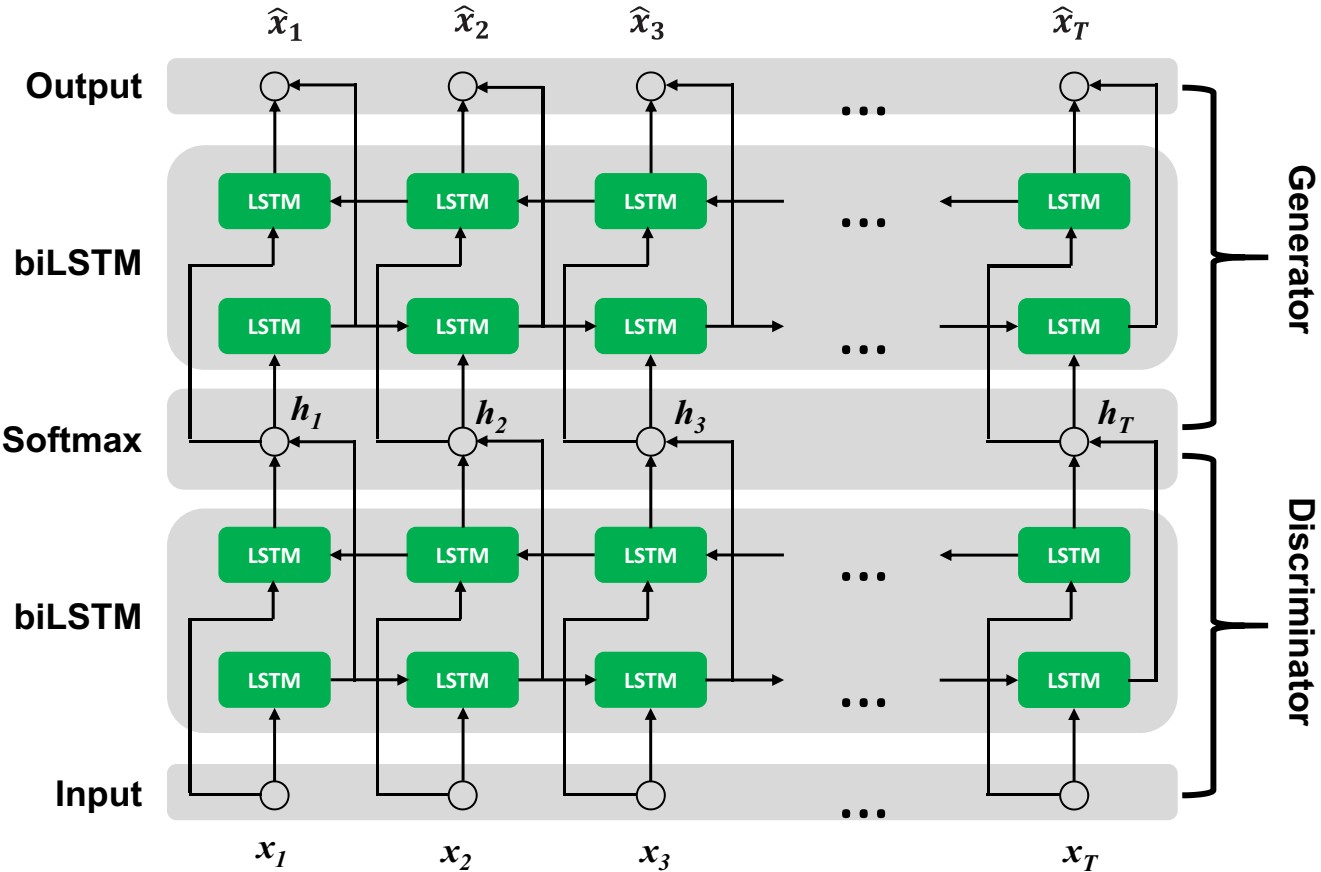

**Fig. 1 The graphical model of combing the discriminator and generator.** The hidden process $h$ is the actual fluorophore number sequence, $x$ and $\hat{x}$ are the measured and predicted fluorescence intensity sequence, respectively.

We checked the classification ability of the discriminator with the synthesized testing subset. The normalized confusion matrixes of the hidden sequence demonstrate the predicted accuracies (Supplementary Fig. 4 for photobleaching event counting).

We then visualized the outputs of the discriminator with two examples of synthesized testing data (Fig. 3(a, b)), in which, Fig. 3 (a) is for photobleaching counting and Fig. 3(b) for dynamic finding. In Fig. 3(a) and Fig. 3(b), the black curves are the synthesized fluorescence intensity traces, the green broken line is the corresponding hidden state paths and the purple broken line is the predicted hidden state paths. For the photobleaching counting (Fig. 3(a)), even for the hidden states with very short duration (such as the state 3 with 2 frames only), our discriminator can give accurate estimates, which is very difficult to do this for the HMM methods. Remarkably, this discriminator is even able to extract hidden state path from the noise distribution, for example, the fluorescence intensity traces between the state 8 and state 5 is slope-like, it is impossible to discern the state paths for human identifier and it is usually excluded with some filters (such as polynomial fitting in our CLDNN[23]). For the dynamic finding (Fig. 3(b)), the discriminator gave almost perfect estimates of the hidden state paths. As expected, the LSTM layers distinguished the actual state paths from the blinking events.

In the photobleaching data analysis (Fig. 2(b)), researchers also care about the predicting accuracy on statistic distribution of bleaching steps, which indicates the protein population. Herein, we tested 1000 synthesized samples (with aSNR 2.98 and equal percentage 10%) as in Fig. 3(c). We achieved a quite good

estimate of the statistic distribution of bleaching steps. The maximum error appeared at the state 7 as about 0.99%.

Further, with the experimental data from monomeric fluorophore Cy5, we checked the DGN net. As expected, there are a vast majority (86.81 ± 0.38%) of monomeric Cy5 molecules bleached with single photobleaching step and only a small fraction (<14%) was bleached with two or more steps because of more Cy5 fluorophores locating within the region of diffraction limit (Fig. 3(d)). This distribution of bleaching step is similar to that of monomeric YFP immobilized on the glass[41].

To check the generalizability of our model, for photobleaching counting, we tested other eight datasets (1000 samples for each dataset) with different SNR (aSNR is 0.67, 0.79, 1.04, 1.35, 1.69, 2.33, 2.98, 3.69, and 4.74, respectively). The black curve (marked as "DGN 10 States") in Fig. 4(a) shows the accuracies under these aSNRs (the accuracy defined as the lowest accuracy of the 10 states as in Supplementary Fig. 4). Even at the noise level aSNR = 1.69, DGN obtained the accuracy of 71.4 ± 4.04 % yet. We also checked the accuracy values of first 5 states. The orange curve (marked as "DGN 5 States") in Fig. 4(a) presents the accuracies under these aSNRs. Under the noise level aSNR = 3.69, we got the accuracy of 94.4 ± 0.31%, which was about the same signal to noise level in our single-molecule imaging data.

For dynamic finding (Fig. 2(b)), we also tested seven datasets (1000 samples for each dataset) with aSNR 0.68, 0.84, 1.23, 1.71, 2.44, 3.84, and 5.45, respectively. The blue curve (marked as "DGN 5 States DF") in Fig. 4(a) shows the accuracies of the 5 states under these aSNRs. For example, at the noise level aSNR = 2.44, DGN obtained the accuracy of 74.0 ± 1.79 % yet.

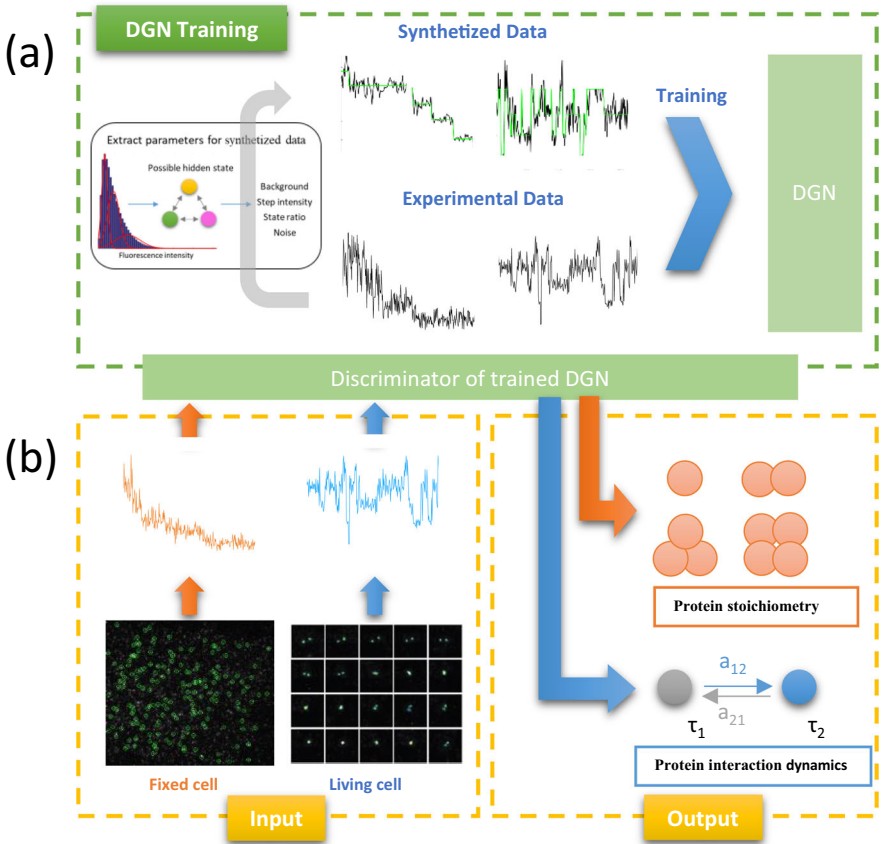

**Fig. 2 The training and performances of DGN on both photobleaching event counting and dynamic finding with the fluorescence intensity traces. a** The training process of DGN with synthesized datasets and experimental datasets. **b** The process of applying DGN to fluorescence traces.

**Comparison with different algorithms**. So far, we have not found any publications of dynamic finding based on fluorescence intensity traces, we focused on the comparison of different algorithms for photobleaching counting.

To compare DGN with other previous methods (such as HMM[3], NoRSE[42], PIF[43], and CLDNN[23] etc.), we synthesized additional test datasets with states from 2 to 5 (because most previous methods can only deal with the questions with maximum state number 5) and aSNRs at 1.18, 1.40, 1.87, 2.40, 2.97, 3.81, and 4.42, respectively (1000 samples for each dataset without zero-step traces)[38]. We retrained the DGN net with hypothetical maximum state number of 10 but tested it with these synthesized test datasets. Figure 4(b) shows the accuracies of these methods under the different aSNRs. From Fig. 4(b), the HMM (blue curve), NoRSE (green curve) and PIF (orange curve) have similar performances with the synthesized test datasets. Our DGN method (black curve) shows much higher accuracies than these three methods with these synthesized testing datasets. For example, when aSNR = 1.40, DGN achieved 79.6% accuracy, but HMM got accuracy 30.9% only. We think the reason is that unlike HMM, biLSTM has long term memory and can extract the information from both past and future context, so even under heavy noise, it can extract the step features from the noise distribution. We also compared our DGN method with our CLDNN method (pink curve) reported previously[23], and found that DGN had similar accuracies with CLDNN. However, the CLDNN has more complicated structure including two convolutional layers and two LSTM layers, in which, the convolutional layers are charge of feature extraction and the LSTM layers discern between the photobleaching and the photoblinking.

The drawback is that the convolutional layers ignore the context information from the time traces hence fail to give full play of LSTM. In DGN method, the biLSTM layer can make full use of the context information, and is able to extract state paths even according to the noise distribution[5,25].

Although most photobleaching event counting is applied to proteins with low subunit number, we think DGN is able to quantify protein oligomers or complex with high-order states. We further tested the performance of DGN to the datasets with much more steps (50 states as in Supplementary Fig. 5). In Supplementary Fig. 5, the aSNR of the 7 datasets are 4.03, 3.22, 2.36, 1.97, 1.56, 1.08, and 0.84, respectively. Under the noise level aSNR 4.03, we can obtain the estimate accuracy 52.62% of the state 49.

**Photobleaching event counting from experimental data**. As in Fig. 2(b), after imaging individual GFP tagged transforming growth factor-β (TGF-β) type II receptors (TβRII) on cell membrane, we applied DGN to analyze 1139 traces (400 frames for each trace) from 21 resting MCF7 cells without TGF-β stimulation, and 1618 traces from 17 MCF7 cells with TGF-β stimulation. TβRII is a crucial component in TGF-β signal transduction, which regulates many important cellular processes, including cell growth, differentiation, migration, and apoptosis[44]. It is believed that TβRII can exist as monomer in resting cells. With the binding of TGF-β, TβRII can interact with TGF-β type I receptor (TβRI) to form the signaling complex (mainly tetramer consisting of two TβRII and two TβRI) and initiate the intracellular signaling[45–47].

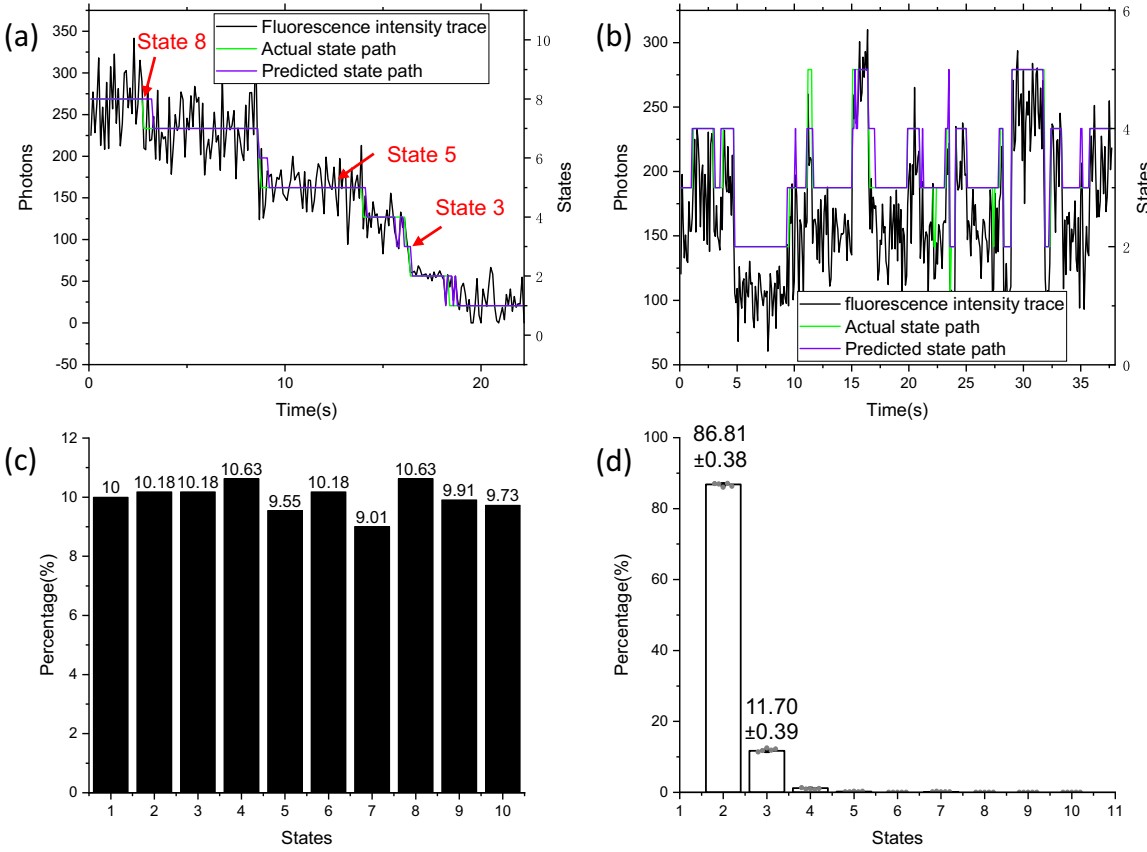

**Fig. 3 Valuating the framework with both synthesized and experimental testing data. a** Predicted results of the pretrained discriminator for photobleaching event counting. **b** Predicted results of the pretrained discriminator for dynamic finding. **c** Estimate of population distribution of the synthesized data. **d** The estimation of the experimental data from monomeric fluorophore Cy5 ($n = 5$ datasets; mean ± SD).

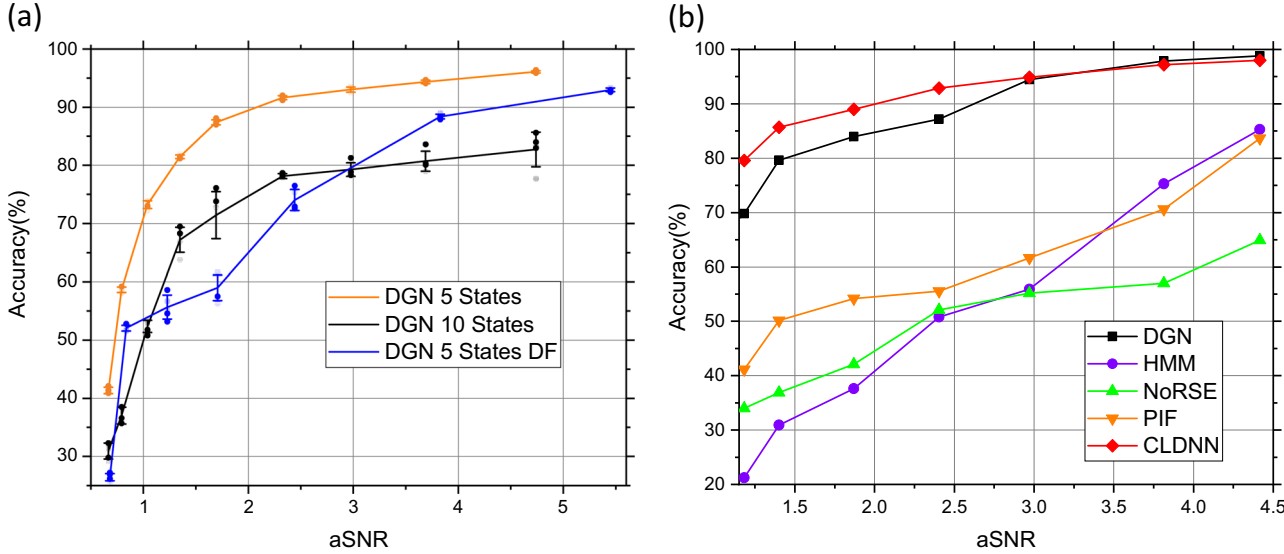

**Fig. 4 The predicting accuracy versus aSNRs. a** The predicting accuracy of DGN for photobleaching counting and dynamic finding ($n = 5$ datasets; Mean ± SD). **b** Comparison with other methods.

In Fig. 5(a), the black curve recorded the photobleaching trace of a single EGFP tagged TβRII molecule that imaged with custom-built total internal reflection fluorescence microscope (TIRFM), and the purple broken line is the predicted state path. Noticeably, there is a transitory state 2 in the trace.

Results from statistical photobleaching steps clearly showed that, in the resting cells, the fraction of TβRII monomers (State 2) and dimers (State 3) is 41.8 ± 0.59 and 38.0 ± 0.87%, respectively, and there are only little trimers and tetramers. After ligand stimulation, monomers undergo severe oligomerization, the

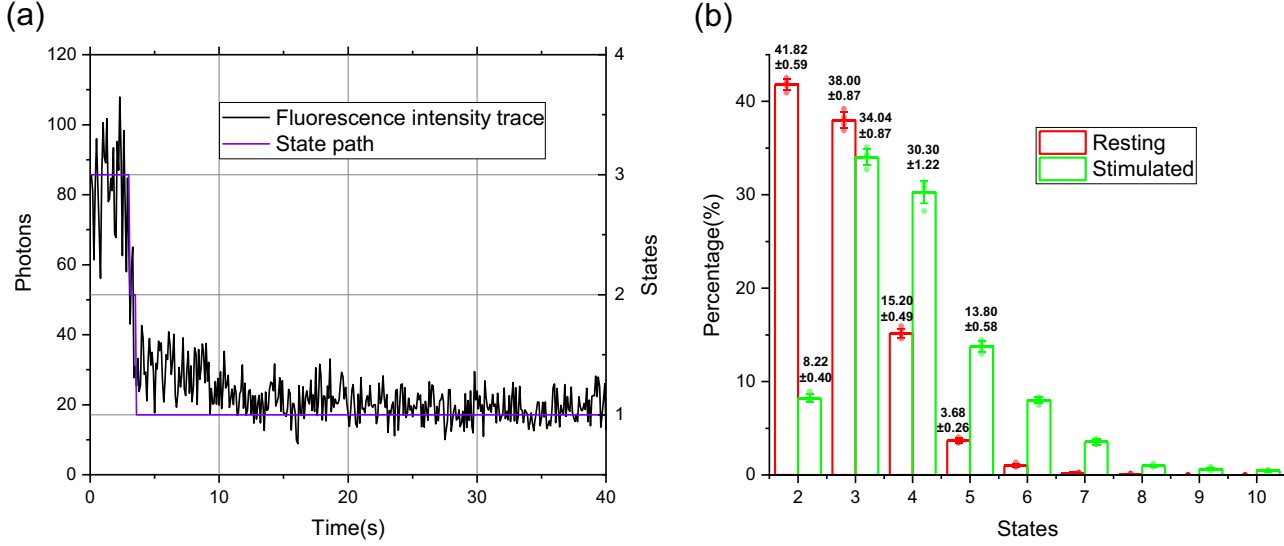

**Fig. 5 The performance of DGN on experimental photobleaching traces. a** An example of the predicted state path with DGN. **b** Predicted state occupancies of every states with DGN ($n = 5$ cell groups; mean ± SD).

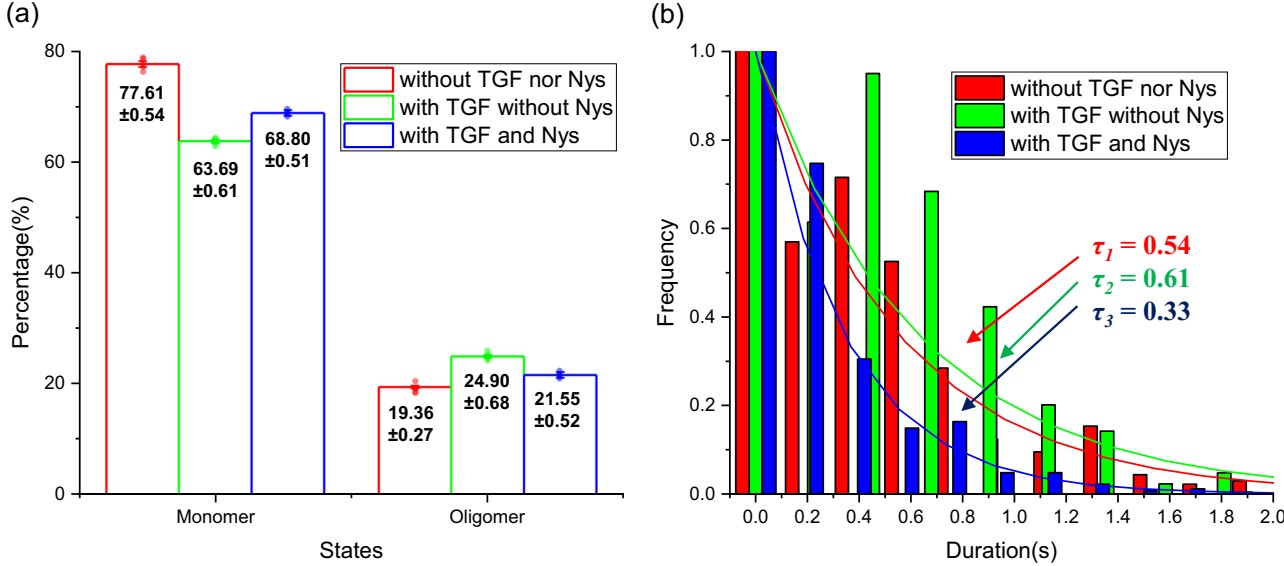

**Fig. 6 The estimated dynamics from experimental fluorescence intensity traces before and after TGF-β (TGF) stimulation, and without and with Nystatin (Nys) treatment. a** The estimated state occupancies of the monomer and oligomer ($n = 5$ cell groups; mean ± SD). **b** The estimated oligomer duration time.

dimers (State 3), trimers (State 4) and tetramers (State 5) increased to $34.0 \pm 0.87$, $30.3 \pm 1.22$ and $13.8 \pm 0.58\%$, respectively, however, the monomers decreased to $8.2 \pm 0.40\%$ (Fig. 5 (b)), suggesting that after the TGF-β stimulation, more monomeric TβRII oligomerize to dimers and other oligomers.

**Dynamic finding from experimental data**. We retrained this framework to analyze single-molecule fluorescence intensity traces of TβRII in living cells (Fig. 2(b)). The fluorescence intensity of observed single molecules is expected to be affected by the association or disassociation process of TβRII, leading to multiple fluorescent states existence.

The analysis of single-molecule TβRII-EGFP traces on Hela cells revealed that TβRII exhibited two dominant fluorescent states. Thus, we referred the low- and high-fluorescent states as

monomer and oligomer (TβRII/TβRII or TβRI/TβRI/TβRII/TβRII), respectively (Fig. 6(a)).

Upon the TGF-β stimulation, the state occupancies of oligomers increased from $19.36 \pm 0.27\%$ (red bar of Fig. 6(a)) to $24.90 \pm 0.68\%$ (green bar of Fig. 6(a)), while that of monomers decreased from $77.61 \pm 0.54$ to $63.69 \pm 0.61\%$. This shows more TβRII have aggregated to form dimers or heterotetramers for signal activation, which is consistent with the previous studies about TβRII signaling mechanism[45,48]. In the meantime, the transition rates (the transition rates ($a_{ij}$) is the probability of transition from state $i$ to state $j$ in unit time) between different aggregation states can be characterized quantitatively as well (Table 1). In resting cells, the transition rate from monomer to dimer or heterotetramer is $a_{12} = 1.47 \pm 0.086 \, \text{s}^{-1}$ and the reversed transition rate is $a_{21} = 1.28 \pm 0.047 \, \text{s}^{-1}$, and after ligand stimulation, they changed to $a_{12} = 1.62 \pm 0.039 \, \text{s}^{-1}$, and $a_{21} =$

**Table 1 State transition rates between monomer and oligomer before and after TGF-β (TGF) stimulation, and without and with Nystatin (Nys) treatment.**

| Cells | $a_{11}$ ($s^{-1}$) | $a_{12}$ ($s^{-1}$) | $a_{21}$ ($s^{-1}$) | $a_{22}$ ($s^{-1}$) | Traces |
|---|---|---|---|---|---|
| Without TGF nor Nys | 18.53 ± 0.086 | 1.47 ± 0.086 | 1.28 ± 0.047 | 18.72 ± 0.047 | 1323 |
| With TGF without Nys | 18.38 ± 0.039 | 1.62 ± 0.039 | 1.03 ± 0.032 | 18.97 ± 0.032 | 5694 |
| With TGF and Nys | 18.10 ± 0.072 | 1.90 ± 0.072 | 1.076 ± 0.28 | 19.92 ± 0.28 | 4459 |

$1.03 \pm 0.032\,s^{-1}$, correspondingly. The increasement of transition rate from monomer to oligomer and the decrement of the reversed transition rate suggest that ligand would drive the balance move more forward to oligomer formation.

After obtained the state paths of all single molecules, we fitted the duration time of the oligomeric state with a single exponential decay and found that the oligomer duration time was prolonged from 0.54 s (red fitting curve of Fig. 6(b)) to 0.61 s (green fitting curve of Fig. 6(b)) after TGF-β stimulation, indicating that the ligand can stabilize the oligomeric state.

In addition, it has also been suggested that lipid-rafts, the cholesterol-rich membrane microdomains, have effect on TGF-β receptor aggregation. By analyzing the diffusion behavior of single-molecule tracking trajectories of TβRI-EGFP, previous research showed that the formation of heteromeric receptor complexes would be hindered when lipid-rafts were disrupted[49]. We also applied our method to analyze 4459 TβRII-EGFP tracking trajectories under similar conditions. The transfected cells were incubated with 50 μg/ml Nystatin for 30 min before the addition of TGF-β, which is able to disrupt the lipid-rafts. Different from the results without Nystatin treatment, the ligand stimulation on the Nystatin treated cells resulted in a smaller increase in the oligomeric component (21.55 ± 0.52%) (blue bar of Fig. 6(a)). The transition rates from monomer to oligomer and in the reverse direction are larger than that of normal cells with TGF-β stimulation (Table 1), but the oligomer duration time is obviously less than that of normal cells with or without TGF-β stimulation. It indicates that disruption of the lipid-rafts can make TGF-β receptor association or disassociation more active, and the oligomers are difficult to stably exist.

## Discussion

In summary, we have designed a discriminator-generator RNN model to analyze the hidden states sequences from the fluorescence intensity traces. This system can be trained in unsupervised manner with the raw experimental time traces, so neither pre-defined state number nor pre-labeling are need. With the biLSTM as the hidden layers of the discriminator and the generator, both the past and future context can be used fully to improve the prediction results. No Markovian hypothesis is embedded in, so it can even be used to treat time sequences of non-Markovian process. It's worth noting that, from the simulation, the biLSTM units can not only reduce the interferences from all types of noise, but also extract step information from the noise distribution.

This system can be used not only for photobleaching event counting, but also for the dynamic finding. We checked its performances on the synthetic test dataset and the experimental dataset of monomeric fluorophore Cy5, and then used it to experimental single-molecule fluorescence intensity traces successfully.

For photobleaching event counting, comparing with HMM, NoRSE and PIF, DGN has outstanding performance because it has longer term memory and can make use the context fully, even extract step information from only the noise distribution, which is impossible for human identifier. Comparing with CLDNN,

DGN can not only count the photobleaching steps but also give the state paths (hence step location), which is very important for calculating the transition rates between polymerization states.

For dynamic finding, DGN estimated all the dynamic properties such as durations of protein association, transition rates during protein interactions and state occupancies of different protein aggregation states, which are very important, as they underlie protein function within cells. We not only find that ligand can drive the balance forward to oligomer formation, but also find that disruption of the lipid-rafts can make TGF-β receptor association or disassociation more active, and oligomers are difficult to stably exist.

## Methods

**Cell culture and transfection.** Full-length human TβRII cDNA were subcloned into the HindIII and BamHI sites of pEGFP-N1 (Clontech), yielding the TβRII-EGFP plasmids[45]. The plasmid was confirmed by DNA sequencing.

HeLa and MCF7 cells were purchased from Cell Resource Center, IBMS, CAMS/PUMC and cultured in Dulbecco's modified Eagle's medium (DMEM, Gibco) supplemented with 10% fetal bovine serum (Hyclone) and antibiotics (50 mg/ml streptomycin, 50 U/ml penicillin) at 37 °C in a 5% $CO_2$ atmosphere. Cells were seeded in a 35-mm glass-bottom dish for 16 h and then transfected with 0.5 μg TβRII-EGFP plasmids in the DMEM medium for 4 h, respectively. Transfection was performed using Lipofectamine 2000 (Invitrogen) according to manufacturer's instructions.

For the ligand stimulation experiments, Cells transfected with TβRII-EGFP were added with 10 ng/ml TGF-β1 (R&D) in phenol red-free DMEM for 15 min at 37 °C before fluorescence imaging.

**Single-molecule imaging.** Single-molecule fluorescence imaging was performed by a home-built TIRF microscope using an inverted Olympus IX71 microscope equipped with a total internal reflective fluorescence illuminator, a 100×/1.45 NA Plan Apochromatic TIRF objective and an electron-multiplying charge-coupled device (EMCCD) camera (Andor iXon DU-897D BV)[46,50]. EGFP molecules were excited by a 488 nm laser at 1 mW (60 W/cm2) (Melles Griot, Carlsbad, CA, USA). The collected fluorescent signals were passed through a filter HQ 525/50 (Chroma Technology), and then directed to the EMCCD camera. The gain of the EMCCD camera was set at 300. Movies of 400 frames were acquired for each sample at a frame rate of 10 Hz.

**Single-molecule tracking with U-Track software.** Time-lapse sequences of single-molecule image were acquired and then tracked with U-Track methods as described in ref. [51] By fitting Gaussian kernels to approximate the two-dimensional point spread function of the microscope objective around local intensity maxima, the sub-pixel localization is achieved. To construct the trajectories, the algorithm first links the detected particles between consecutive frames, and then links the generated track segments to simultaneously close gaps and capture particle merging and splitting events.

**Statistics and reproducibility.** Bootstrap method was used to ensure the statistics and reproducibility.

To estimate the state distribution of the experimental data from monomeric fluorophore Cy5, five sub-datasets were generated by randomly extracting 72% of the total 2224 trajectories, the means and standard deviations were obtained as in Fig. 3(c), in which, the standard deviations from the bootstrap analysis are typically <0.5%.

To check the generalizability for photobleaching counting, we tested nine datasets (1000 samples for each dataset) with different SNR (aSNR is 0.67, 0.79, 1.04, 1.35, 1.69, 2.33, 2.98, 3.69, and 4.74, respectively). For each dataset, five sub-datasets were generated by randomly extracting 72% of the total trajectories, the means and standard deviations of accuracy were obtained as in Fig. 4(a).

To analyze the state distributions before and after TGF-β stimulation, five cell groups were generated randomly from the total experimental cells, the means and standard deviations were obtained as in Fig. 5(b).

To analyze the state distributions in live cells, five cell groups were generated randomly as well from the total experimental cells, the means and standard deviations were obtained as in Fig. 6(a).

**Reporting summary**. Further information on research design is available in the Nature Research Reporting Summary linked to this article.

## Data availability

All the data of this manuscript are archived in the public repositories Dryad (https://doi.org/10.5061/dryad.4qrfj6q64) by the link: https://datadryad.org/stash/share/9liw8QzlGcUgwdIz5nahbp6wVjXhZvD03ZKp_w5bURg.[38]

## Code availability

The custom codes described in the paper is deposited in the public repositories Zenodo (https://doi.org/10.5281/zenodo.4030065) by the link: https://zenodo.org/record/4030065#.X2Bj_WgzYuU.[24]

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

## Acknowledgements

This work was supported by the National Natural Science Foundation of China (22077124, 21735006, 91939301) and the Chinese Academy of Sciences.

## Author contributions

J.Y. designed the study, conducted analysis, curated the data, formulated methodology, and wrote the paper. R.Z., J.X., M.C., Z.Q., and X.K. performed the single-molecule imaging and tracking experiments. X.F. supervised the project, and reviewed and edited the paper.

## Competing interests

The authors declare no competing interests.
