## [Peer Review File · Communications Biology]

Reviewers' comments:

Reviewer #1 (Remarks to the Author):

In this manuscript the authors presented a deep learning approach to analyze fluorescence traces for photobleaching event counting and dynamic analysis of membrane protein interactions. The traditional methods of solving such problems based on signal processing, such as the hidden Markov model (HMM), are prone to low signal-to-noise ratio and require non-intuitive parameter tuning. Deep neural networks (DNN) provide an alternative approach.

However, after careful evaluation, the reviewer has several concerns with the experiment design and model validation, which make the current manuscript not suitable for publication in its current form.

- The authors claimed "entirely unsupervised training" capability several times throughout the manuscript, however only showed results with supervised pre-trained models. To support the authors' claim, the reviewer would like to see quantitative answers of following questions:
 - (a) How does the modeling accuracy of synthesized pre-training data affect the final DNN model performance?
 - (b) Can a DNN model be successfully trained in unsupervised manner, i.e., without supervised pre-training?
 - (c) If so, how is the prediction accuracy compared against models with supervised pre-training?

- The DNN models reported in this manuscript were first (1) pre-trained with synthesized data, (2) refined with experimental data, and then (3) tested with synthesized data to quantify prediction accuracy. This workflow is somehow confusing, as the reviewer wants to understand:
 - (a) Since the testing was performed with synthesized data, why was step (2) necessary? Would the pre-trained model from step (1) yield better results?
 - (b) How can one validate that the DNN model indeed learned information during step (2)?
 - (c) Can the testing accuracy with synthesized data represent accuracy with experimental data?

- With a close look at the training curves (Fig. S3), the reviewer wants to ask:
 - (a) The training loss is significantly lower than validation loss, does it mean overfitting?
 - (b) The validation loss is slightly decreasing within the first 3 epochs. How many epochs was the DNN trained? How was the model selected?

- The hidden state vector h in the network model is assumed to be the fluorophore number. While this is true for supervised learning, there is not a guarantee for unsupervised learning process that h , as hidden states within the network, will converge to the fluorophore number. How did the authors address this problem?

- Besides, more introduction to dynamic finding will be helpful to readers outside the field. Especially the differences between the signals of photobleaching event counting and dynamic finding, and the current state-of-the-art techniques employed by researchers.

Reviewer #2 (Remarks to the Author):

The authors tackle the problem of analyzing 1D fluorescence intensity time traces to determine the underlying hidden fluorophore states, which reveal information about photobleaching events and dynamics. Their proposed approach, DGN uses a discriminator-generator network (DGN) architecture which they show on simulated data achieves similar photobleaching event counting performance as their previous CLDNN network (a hybrid convolutional LSTM network) but with the promise of interpretable latent hidden states which can reveal dynamics information, and with the

claim that the DGN is unsupervised whereas CLDNN requires supervision with large training datasets. I find the ideas proposed interesting, and determining hidden dynamics states seems like a useful application, however, I'm not quite convinced that the manuscript fully supports some of the claims made.

Major points:

- * The author's reference previous work on GANs, but do not mention any related work on LSTM autoencoders or variational autoencoders (VAEs). Please include a discussion of this work and the your proposed work in the context of this work.

- * The challenges in all of these types of VAE models, is how to formulate appropriate architectures, training loss functions and datasets for training to ensure the latent space of hidden states of the model are interpretable and useful (rather than, say, some random noise space as in GANs). Particularly in the unsupervised case, this is quite a difficult challenge. The authors reveal that they used a "trick" (line 200) that they actually pretrain the two halves of their DGN network separately using supervised learning on synthetic data first. After that, they train the whole system in an unsupervised way. So this leads me to ask the following questions, none of which I believe are addressed in the current manuscript-- what's the DGN performance after just the supervised training (no unsupervised training)? What value does the unsupervised training add? How closely must the distributions of data inputs match between the supervised and unsupervised training?

- * I think it's a bit of a stretch to call the approach "unsupervised" if large amounts of supervised synthetic training data matching the experimental data distribution are required. I'd be happier to accept the "unsupervised" label if (1) the method still works ****without**** the supervised training step or (2) the additional training with the unsupervised data significantly improves the accuracy beyond just what the "supervised" model can achieve

- * For all of the model evaluation and comparison with other algorithms sections, line 226-337, there is a lot of discussion about using different "datasets" for evaluation -- were all of these synthetic datasets? Again, as above, how closely were the distributions of supervised and unsupervised training data matched in each case? I'd really like to see how much value the additional unsupervised training adds onto the DGN just trained with supervision.

- * lines 232-336 - the figure referred to appears to show both training losses decreasing significantly, but both validation losses appear to drop only 1/100th the amount the training losses drop; technically this is "going down uniformly" but my interpretation of this figure is that the validation loss isn't really changing at all -- could you please explain or investigate this? how is the validation data different from the training data?

- * As the DGN has similar performance as the authors' previous CLDNN work, I'd like to see a stronger justification of the merits of the DGN beyond the description in lines 323-330; in particular, I want experiments (ideally synthetic and experimental) showing it can extract state paths on data from the unsupervised data distribution.

- * even if all of the above are addressed, the shape of the paper is fundamentally evaluating the model on synthetic data, in which the hidden states are known and accuracy can be evaluated, but then in lines 364+, all that's done on real experimental data is revealing the model's predictions, without any quantification or verification that they might be correct. I understand it's not easy to directly know the ground truth real hidden states to be able to evaluate the work -- but the work could be much stronger if somehow the hidden states determined on real experimental data could be somehow validated.

Minor points:

- * The manuscript is structured well and fairly clearly outlines the ideas and flows well. However, I'd recommend the manuscript be screened by an editing service for improved use of English language. For example, "severally" is used in two places where I think it's meant to be "separately". "batch 4" > "batch size 4". This would improve the readability.

- * please include legends in all plots, e.g. Fig 5a, etc.

- * some of the language is a little strong -- line 419 "entirely" unsupervised and line 424 "miraculously" ... "eliminate"... "noise" lines 424-426. and some other places.

Response to Reviewer's Comments

Reviewer #1:

In this manuscript the authors presented a deep learning approach to analyze fluorescence traces for photobleaching event counting and dynamic analysis of membrane protein interactions. The traditional methods of solving such problems based on signal processing, such as the hidden Markov model (HMM), are prone to low signal-to-noise ratio and require non-intuitive parameter tuning. Deep neural networks (DNN) provide an alternative approach.

However, after careful evaluation, the reviewer has several concerns with the experiment design and model validation, which make the current manuscript not suitable for publication in its current form.

The authors claimed “entirely unsupervised training” capability several times throughout the manuscript, however only showed results with supervised pre-trained models. To support the authors’ claim, the reviewer would like to see quantitative answers of following questions:

(a) How does the modeling accuracy of synthesized pre-training data affect the final DNN model performance?

Answer: Thanks for reviewer’s suggestion. We have removed the word “entirely” in this revised manuscript. In this revised manuscript, we trained the DGN with new strategy. The exact hidden states under the experimental traces are unknown, unsupervised training the net just with the experimental data cannot guarantee convergence, so we synthesized the datasets referring to the experimental ones. We used the synthesized data in two purposes, introducing the net convergence and validating the net in this revised paper. Anyway, we can still estimate the performance by checking the population distribution and the state paths of the experimental traces as in figure R1.

Figure R1. The performance on the experimental data from monomeric fluorophore Cy5 after just the supervised training. (a) the estimated state paths, the state 2 was not found. (b) the estimated state paths, there was a wrong state 9. (c) the estimated population distribution. The population percentage of state 9 is obvious unreasonable.

(b) Can a DNN model be successfully trained in unsupervised manner, i.e., without supervised pre-training?

Answer: We trained the DGN with new strategy in this revised manuscript. As the explanation in this revised main text (section “Training the network”), just as training GANs, training our DGN is unstable as well because the loss function cannot ensure convergence.

Actually, to train the net in unsupervised manner without supervised pre-training, we had tried to remould the loss function with variational inference methods, but found that the performance is heavily dependent on the pre-defined maximum possible state number. In figure R2, we used the variational Bayes expectation maximization (VBEM) method (Pattern recognition and machine learning. Cristopher M. Bishop. Information Science and Statistics. Springer 2006) to estimate state number from the TGF- β type II receptors datasets, when we pre-defined the maximum possible state number as 6, we got the estimated state number 5 (left panel), however, when the maximum possible state number as 10, we got the estimated state number 8 (right panel). We think that the bad performance of VBEM is due to the VBEM method is still model based method and its parameters may not be able to describe the feature under the data fully. However, the neural networks can approach any statistical model and get satisfying performance.

Figure R2. Estimating the state number with VBEM method. Left panel is the result with pre-defined maximum possible state number as 6 and right panel is the result with pre-defined maximum possible state number as 10.

(c) If so, how is the prediction accuracy compared against models with supervised pre-training?

Answer: We compared the performance of DGN under current training strategy with the supervised pre-training as well and found the prediction accuracy is similar but current training strategy needs more epochs for convergence.

The DNN models reported in this manuscript were first (1) pre-trained with synthesized data, (2) refined with experimental data, and then (3) tested with synthesized data to quantify prediction accuracy. This workflow is somehow confusing, as the reviewer wants to understand:

(a) Since the testing was performed with synthesized data, why was step (2) necessary? Would the pre-trained model from step (1) yield better results?

Answer: We have characterized the net with both the synthesized data and experimental data from monomeric fluorophore Cy5. Because the hidden states under the experimental data are unknown, we had to characterize the accuracy and population distribution with the synthesized data. Anyway, the prediction results of experimental data from monomeric fluorophore Cy5 are reasonable and proved the performance of the net indirectly.

The reason for training our framework with experimental data is that although we designed the synthesized datasets to simulate the actual experimental data, it may be still hard to include all the features of the real photobleaching traces. Please refer to figure R1 as well.

(b) How can one validate that the DNN model indeed learned information during step (2)?

Answer: To answer this question, we can compare figure R1(b) with figure 3(c) of main text. After training only with synthetic data, the prediction of the population distribution for the experimental data is unreasonable in figure R1(b).

(c) Can the testing accuracy with synthesized data represent accuracy with experimental data?

Answer: We think that the testing accuracy with synthesized data cannot exactly and fully represent the accuracy of the experimental data. However, as in the answer (a), we cannot know the exact and full features of the experimental data, we had to simulate the experimental data with synthesized one, and to check the performance with the synthesized data. To make up for this defect, we also tested this net with the experimental data from monomeric fluorophore Cy5, although these proofs were indirect and insufficient.

With a close look at the training curves (Fig. S3), the reviewer wants to ask:

(a) The training loss is significantly lower than validation loss, does it mean overfitting?

Answer: We judged whether overfitting start to emerge by checking the training loss monotonically decrease but the validation loss starts to increase.

(b) The validation loss is slightly decreasing within the first 3 epochs. How many epochs was the DNN trained? How was the model selected?

Answer: In the last manuscript, the epochs number is 3 for all the unsupervised training. In this revised manuscript, with new training strategy, much more epochs have been used as in figure S3.

The hidden state vector h in the network model is assumed to be the fluorophore number. While this is true for supervised learning, there is not a guarantee for unsupervised learning process that h , as hidden states within the network, will converge to the fluorophore number. How did the authors address this problem?

Answer: Yes, training this net only with the experimental data cannot guarantee convergent to the true fluorophore number. We have tried to amend the loss function referring to variational Bayes methods. However, we finally found that the variational Bayes expectation maximization (VBEM) method cannot give stable results. We think the reason behind it is that no model can simulate the real distribution of the experimental data exactly. So we diverted our attention to new training strategy, namely, introducing the net convergence with the synthetic data.

Besides, more introduction to dynamic finding will be helpful to readers outside the field. Especially the differences between the signals of photobleaching event counting and dynamic finding, and the current state-of-the-art techniques employed by researchers.

Answer: we add the sentence "The main difference of dynamic finding from photobleaching event counting is that the dynamic finding is to track the processes of molecule association and dissociation, which was presented as

the molecule diffusion state changes or fluorescence intensity jumps” in the first paragraph to introduce the dynamic finding.

We introduce the photobleaching event counting, dynamic finding, and current state-of-the-art techniques in the introduction section.

Reviewer #2:

The authors tackle the problem of analyzing 1D fluorescence intensity time traces to determine the underlying hidden fluorophore states, which reveal information about photobleaching events and dynamics. Their proposed approach, DGN uses a discriminator-generator network (DGN) architecture which they show on simulated data achieves similar photobleaching event counting performance as their previous CLDNN network (a hybrid convolutional LSTM network) but with the promise of interpretable latent hidden states which can reveal dynamics information, and with the claim that the DGN is unsupervised whereas CLDNN requires supervision with large training datasets. I find the ideas proposed interesting, and determining hidden dynamics states seems like a useful application, however, I'm not quite convinced that the manuscript fully supports some of the claims made.

Major points:

The author's reference previous work on GANs, but do not mention any related work on LSTM autoencoders or variational autoencoders (VAEs). Please include a discussion of this work and the your proposed work in the context of this work.

Answer: I introduced and discussed the auto-encoding variational Bayes (VAEs) method in the introduction of this revised manuscript.

The challenges in all of these types of VAE models, is how to formulate appropriate architectures, training loss functions and datasets for training to ensure the latent space of hidden states of the model are interpretable and useful (rather than, say, some random noise space as in GANs). Particularly in

the unsupervised case, this is quite a difficult challenge. The authors reveal that they used a "trick" (line 200) that they actually pretrain the two halves of their DGN network separately using supervised learning on synthetic data first. After that, they train the whole system in an unsupervised way. So this leads me to ask the following questions, none of which I believe are addressed in the current manuscript-- what's the DGN performance after just the supervised training (no unsupervised training)? What value does the unsupervised training add? How closely must the distributions of data inputs match between the supervised and unsupervised training?

Answer: After the supervised training only with the synthetic data, we can get satisfying performance on the synthetic data, but there are many dissatisfactory results on the experimental data such as in figure R1. In figure R1 (a), the state 2 was not found, and in figure R1 (b), there was a wrong state 9. In figure R1(c), the estimated population percentage of state 9 is obvious unreasonable. We think that it is hard to include all the features of the real photobleaching traces with the synthetic ones, so the value of unsupervised training is to train the net extract special features of the experimental data, meanwhile guarantee the net convergence in the direction of supervised training.

When we synthesized the simulating data, although the Poisson distributed shot noise, the Gaussian distributed random noise, and the fluorescence blinking have been taken into count, we were afraid of other implied features yet.

Figure R1. The performance on the experimental data from monomeric fluorophore Cy5 after just the supervised training. (a) the estimated state paths, the state 2 was not found. (b) the

estimated state paths, there was a wrong state 9. (c) the estimated population distribution. The population percentage of state 9 is obvious unreasonable.

I think it's a bit of a stretch to call the approach "unsupervised" if large amounts of supervised synthetic training data matching the experimental data distribution are required. I'd be happier to accept the "unsupervised" label if (1) the method still works ****without**** the supervised training step or (2) the additional training with the unsupervised data significantly improves the accuracy beyond just what the "supervised" model can achieve.

Answer: Thanks to reviewer's suggestion. In this revised manuscript, we have amended the training strategy make it more unsupervised.

Actually, for entire unsupervised training, we had tried to remould the loss function with variational inference methods, but found that the performance is heavily dependent on the pre-defined maximum possible state number. In figure R2, we used the variational Bayes expectation maximization (VBEM) method (Pattern recognition and machine learning. Cristopher M. Bishop. Information Science and Statistics. Springer 2006) to estimate state number from the TGF- β type II receptors datasets, when we pre-defined the maximum possible state number as 6, we got the estimated state number 5 (left panel), contrastively, when the maximum possible state number 10, we got the estimated state number 8 (right panel),

We think that the bad performance is due to the VBEM method is still model based method and its parameters may not be able to describe the feature under the data fully. However, the neural networks can approach any statistical model and get satisfying performance.

Figure R2. Estimating the state number with VBEM method. Left panel is the result with pre-defined maximum possible state number as 6 and right panel is the result with pre-defined maximum possible state number as 10.

For all of the model evaluation and comparison with other algorithms sections, line 226-337, there is a lot of discussion about using different "datasets" for evaluation -- were all of these synthetic datasets? Again, as above, how closely were the distributions of supervised and unsupervised training data matched in each case? I'd really like to see how much value the additional unsupervised training adds onto the DGN just trained with supervision.

Answer: We have characterized the net with both the synthesized data and actual experimental data from monomeric fluorophore Cy5.

Because the hidden states under the experimental data are unknown, which cannot be used to characterize the accuracy and population distribution. Anyway, the prediction results of experimental data from monomeric fluorophore Cy5 are reasonable and proved the performance of the net indirectly.

As in the answer above, the training with experimental data improved the performance of the net indeed.

lines 232-336 - the figure referred to appears to show both training losses decreasing significantly, but both validation losses appear to drop only 1/100th

the amount the training losses drop; technically this is "going down uniformly" but my interpretation of this figure is that the validation loss isn't really changing at all -- could you please explain or investigate this? how is the validation data different from the training data?

Answer: Statistically, there is no difference between the training data and validation data. The loss curves are for unsupervised training in the last manuscript. Because the convergence of the supervised training had finished, so the loss of this unsupervised training drops unobviously. In this revised manuscript, with new training strategy, much more epochs had to be used as in figure S3.

As the DGN has similar performance as the authors' previous CLDNN work, I'd like to see a stronger justification of the merits of the DGN beyond the description in lines 323-330; in particular, I want experiments (ideally synthetic and experimental) showing it can extract state paths on data from the unsupervised data distribution.

Answer: In figure 4(b), we showed that DGN has similar accuracy as CLDNN. However, as described in the introduction section, CLDNN is supervised learning framework but DGN is unsupervised one. CLDNN includes some convolutional layers for extracting step-like features and LSTM layers distinguish between photobleaching and photoblinking. It is the convolutional layers that causes it is difficult to get the state paths.

Figure 5(a) shows an example of actual experimental trace, and the estimated state path with our DGN.

even if all of the above are addressed, the shape of the paper is fundamentally evaluating the model on synthetic data, in which the hidden states are known and accuracy can be evaluated, but then in lines 364+, all that's done on real experimental data is revealing the model's predictions, without any quantification or verification that they might be correct. I

understand it's not easy to directly know the ground truth real hidden states to be able to evaluate the work -- but the work could be much stronger if somehow the hidden states determined on real experimental data could be somehow validated.

Answer: Thanks to the reviewer. We have characterized the net with both the synthesized data and experimental data from monomeric fluorophore Cy5.

As reviewer's words, "it's not easy to directly know the ground truth real hidden states to be able to evaluate the work", we had to characterize the accuracy and population distribution with the synthesized data. Anyway, the prediction results of actual experimental data from monomeric fluorophore Cy5 are reasonable and proved the performance of the net indirectly.

Minor points:

The manuscript is structured well and fairly clearly outlines the ideas and flows well. However, I'd recommend the manuscript be screened by an editing service for improved use of English language. For example, "severally" is used in two places where I think it's meant to be "separately". "batch 4" > "batch size 4". This would improve the readability.

Answer: We have revised these mistakes in this revised manuscript.

please include legends in all plots, e.g. Fig 5a, etc.

Answer: We have added the legends in this revised manuscript.

some of the language is a little strong -- line 419 "entirely" unsupervised and line 424 "miraculously" ... "eliminate"... "noise" lines 424-426. and some other places.

Answer: We have corrected these words as following:

1. Delete "entirely";
2. Miraculously -> Remarkably;

3. eliminate -> reduce.